# Smartphone- and Paper-Based Delivery of Balance Intervention for Older Adults Are Equally Effective, Enjoyable, and of High Fidelity: A Randomized Controlled Trial

**DOI:** 10.3390/s23177451

**Published:** 2023-08-27

**Authors:** Vipul Lugade, Molly Torbitt, Suzanne R. O’Brien, Patima Silsupadol

**Affiliations:** 1Division of Physical Therapy, Decker College of Nursing and Health Sciences, SUNY Binghamton University, 4400 Vestal Parkway East, Binghamton, New York, NY 13902, USA; torbittm@upstate.edu (M.T.); sobrien@binghamton.edu (S.R.O.); patimas@binghamton.edu (P.S.); 2Department of Physical Therapy Education, College of Health Professions, SUNY Upstate Medical University, 750 East Adams Street, Syracuse, NY 13210, USA

**Keywords:** gait, posture, remote monitoring, older adults, home-based training

## Abstract

Home-based rehabilitation programs for older adults have demonstrated effectiveness, desirability, and reduced burden. However, the feasibility and effectiveness of balance-intervention training delivered through traditional paper-versus novel smartphone-based methods is unknown. Therefore, the purpose of this study was to evaluate if a home-based balance-intervention program could equally improve balance performance when delivered via smartphone or paper among adults over the age of 65. A total of 31 older adults were randomized into either a paper or phone group and completed a 4-week asynchronous self-guided balance intervention across 12 sessions for approximately 30 min per session. Baseline, 4-week, and 8-week walking and standing balance evaluations were performed, with exercise duration and adherence recorded. Additional self-reported measures were collected regarding the enjoyment, usability, difficulty, and length of the exercise program. Twenty-nine participants completed the balance program and three assessments, with no group differences found for any outcome measure. Older adults demonstrated an approximately 0.06 m/s faster gait velocity and modified balance strategies during walking and standing conditions following the intervention protocol. Participants further self-reported similar enjoyment, difficulty, and exercise effectiveness. Results of this study demonstrated the potential to safely deliver home-based interventions as well as the feasibility and effectiveness of delivering balance intervention through a smartphone-based application.

## 1. Introduction

Aging is accompanied by deterioration in physical, cognitive, and psychological effects [1,2]. Concurrent with age-related deterioration, falls remain a major health concern for older adults, with approximately one-in-three older adults reporting a fall at least once a year [3,4]. As the number of adults over the age of 65 is expected to exceed 21.6% of the American population by 2040, [5] it is vital to develop programs to address age-related decline in balance ability in order to improve older adult quality of life and reduce the healthcare burden.

Balance-intervention studies for older adults have demonstrated effectiveness in improving single- and dual-task balance performance [6,7]. While most intervention studies have been performed in controlled clinical or laboratory settings, home-based training programs have been shown to be effective, feasible, and more desirable [8,9,10]. Home-based training programs for older adults are often delivered via instruction by physical therapists or rehabilitation experts and through the medium of paper for instruction and logging of activity [11]. Given the increased ubiquity of smartphones, even among older populations, it is unknown how feasible and effective a smartphone-delivered balance-intervention program can be among older adults.

Therefore, the overall objective of this study was to evaluate if a home-based 4-week balance-intervention program could equally improve balance performance when delivered via smartphone or paper among older adults. We hypothesized that novel smartphone-delivered interventions could be successfully performed by older adults (based on compliance, fidelity, and the time required to complete home-based exercise programs), compared to traditional paper-based interventions. We further hypothesized that, regardless of the delivery method, all adults receiving intervention would demonstrate increased gait velocity and improved balance performance (i.e., greater dynamic stability as measured by the center of mass and base of support interaction as well as using appropriate postural strategies) under walking and standing conditions, respectively. Finally, we hypothesized that individuals would demonstrate a short-term retention effect, with increased gait velocity and standing balance up to 4-weeks post intervention.

## 2. Materials and Methods

Older adults were recruited through flyers posted in the surrounding senior and community centers. Inclusion criteria included being over the age of 65 years, having the ability to walk at least 10 m without the use of an assistive device, demonstrating intact cognitive function based on scoring 18/22 or greater on the Montreal Cognitive Assessment—Blind [12], and having access to an Android or iOS smartphone. Participants were excluded if they self-reported any lower limb amputation, visual impairment uncorrectable with lenses, uncontrolled hypertension or diabetes, neurological or musculoskeletal impairment, or demonstrated persistent symptoms of dizziness or lightheadedness. All procedures were approved by Binghamton University’s Institutional Review Board, with written informed consent obtained from all participants prior to enrollment into the study. The study was registered in the ClinicalTrials.gov registry (ID: NCT05140044).

### 2.1. Sample Size

The sample size was calculated based on our previous training study [11]. Utilizing a power of 0.8, an effect size of 0.27, and a 0.05 alpha level for changes in gait velocity over three repeated measurements among two groups, the estimated sample size was 24 participants. In order to account for attrition, we attempted to recruit at least 30 participants into the study.

### 2.2. Procedures

Eligible participants were asked to complete initial, four-week, and eight-week follow-up evaluations in the laboratory. During these visits, participants completed clinical, balance, and gait activities under the supervision of a trained investigator (SRO). During the initial visit, participants were asked to complete questionnaires in regards to their fall history, smartphone usage, and medications. During all visits, assessments of depression and perception of balance ability were performed using the Geriatric Depression Scale (GDS) [13] and the Activities-Specific Balance Confidence (ABC) Scale [14], respectively. Participants were also asked to complete two trials of the Timed Up and Go Test (TUG) [15] during each visit.

Participants were then asked to stand quietly in a computerized dynamic posturography system, with immersive virtual environments and a sway referenced force plate (Bertec Inc., Columbus, OH, USA) for 6 postural conditions to complete the sensory organization test (SOT). Each condition consisted of up to three 20 s trials. For condition 1, participants were asked to stand quietly on the force plate with their eyes open. For condition 2, participants were asked to stand quietly with their eyes closed. For condition 3, participants were asked to stand quietly with their eyes open, with the visual surround changing based on their postural sway. For conditions 4–6, participants stood while the force plate swayed at a 1:1 ratio in reference to their body sway. For condition 4, eyes were open; for condition 5 eyes were closed; for condition 6, eyes were open, but the visual surrounding also changed based on their postural sway. During all standing trials in the posturography system, a fall-arrest harness was placed on participants, with a trained investigator present to ensure safety. The equilibrium scores (ES) across each of the six conditions were automatically computed, with scores closer to 0 indicating increased anteroposterior sway and loss of balance [16]. The sway area and total path excursion were further computed from the center of pressure trajectories during each standing condition, as previously described [17].

Subsequently, thirty-nine reflective markers were placed on bony landmarks of the body, following the Vicon full-body plug-in gait model. Three-dimensional marker trajectories were captured using a 10-camera motion-analysis system (Vicon Motion Systems, Centennial, CO, USA) at 100Hz. Participants were asked to ambulate at a self-selected comfortable speed across a 10 m walkway under single- and dual-task conditions. Since an impaired ability to maintain balance while simultaneously performing cognitive tasks is associated with an increased risk of falling [18,19], participants performed a concurrent verbal fluency task during dual-task walking conditions. For the verbal fluency while walking task, participants were asked to respond with as many words as they could which started with a randomly selected letter of “F”, “A”, or “S”, while also walking at their comfortable self-selected speed. Three walking trials were performed for each condition. All data collections were conducted by two investigators (MT and PS), who were otherwise blinded to participant group assignment.

Spatiotemporal and balance measures during gait were determined across all walking conditions. The whole-body center of mass (CoM) was computed based on the weighted sum of a 13-segment model and anthropometric measures [20]. Gait velocity (GV) was based on the quotient of the center-of-mass displacement and time during steady-state walking, with step width computed from the mediolateral displacement between ankle joint centers at each heel-strike event. The displacement of the extrapolated center of mass (XcoM) [21] in relation to the base of support (BoS) [20] was further calculated as a measure of balance during ambulation. Briefly, the XcoM was computed as:$$\mathrm{XcoM}=\mathrm{CoM}+\frac{\mathrm{CoMv}}{\sqrt{g/l}}$$
where CoMv is the CoM velocity; *l* is the sagittal plane distance from the CoM to the ankle; *g* is the acceleration due to gravity. The BoS was computed based on markers placed on the feet as well as directly measured foot widths and foot lengths, as previously described [22,22]. The XcoM-BoS displacement, or margin of stability, was reported at the moment of toe-off, as this has been shown to be the best approximation of the margin of stability during ambulation [23].

### 2.3. Intervention

Following the first visit assessments, participants were randomly assigned to one of two intervention groups, smartphone or paper, based on a computer-generated list. The allocation sequence and the following intervention protocol were provided by a trained Physical Therapist (SRO) who was blinded to participant performance across all three visits to the laboratory. The trainer provided participants with appropriate intervention procedures, with the exercise routine enabled on participant smartphones for those individuals randomized to the smartphone. Participants were blinded to the existence of the alternate exercise delivery methodology.

All participants were asked to complete a twelve-session training program, with approximately 30 min of activity per session, three times a week for four weeks in their homes. Across the 12 sessions, participants received balance training following Gentile’s taxonomy of movement tasks [24], progressing from stance activities, stance activities with hand manipulation, transitional activities, gait activities, and finally gait activities with hand manipulation (Table 1). Along with single-task balance activities, dual-task motor–motor tasks, such as backward tightrope-walking with arm alternation were interspersed across the intervention program. Older adults in the paper instruction group received four exercise booklets, with detailed written instructions and pictures of each exercise that they needed to perform across the four-week period (Figure 1). Both instruction methods included the number of repetitions and sets to be completed for each exercise. Participants in the phone-based group were provided with identical instruction which were delivered through the Improve application (Figure 2). The application, which was installed on each participants’ iOS or Android smartphone, provided participants with the ability to track exercises, review workout instructions, as well as start and pause exercises. Upon completion of three sessions of each weeks’ exercise program, the following week’s program was enabled, with the prior week disabled. Upon starting each exercise program, the application saved data regarding when each exercise was started and stopped in order to compile compliance and total time to completion for each participant. Participants in the traditional paper-based group were asked to log the start and end time of each exercise session. All participants performed the exact same exercises in the order prescribed on their self-selected days of the week and at their preferred time of day, with all older adults contacted by the trainer (SRO) on a weekly basis during the course of the four-week training program to ensure safety and compliance. Participants were also asked to perform all tasks in a comfortable, everyday environment. Finally, all participants were instructed to continue any other normal pattern of activity and not to begin any new physical activity programs during the course of the 8-week study.

Upon completion of the four-week intervention, participants were provided with a questionnaire during their second visit to the laboratory regarding: (1) how effective the exercises were in improving balance on a 7-point Likert Scale, where 0 represented “Not at all effective” to 6 indicating “Very effective”; (2) how much they enjoyed the exercises on a 7-point Likert Scale, where 0 represented “Did not enjoy at all” to 6 being “Enjoyed tremendously”; (3) the difficulty of the intervention on a 5-point scale with each point corresponding to “very difficult”, “difficult”, “fair”, “easy”, and “very easy”; (4) the length of the exercise programs on a 5-point scale indicating “very long”, “long”, “just right”, “short”, and “very short”. Furthermore, among those in the smartphone group, 7-point scales were used to ask questions regarding whether using a smartphone application allowed the participant to “accomplish tasks more quickly”, “improved exercise performance”, “made it easier to exercise”, and were “useful in my exercise” in comparison to traditional methods, where 0 indicated “Extremely Untrue”, 3 indicated “Neither”, and 6 being “Extremely True”.

## 3. Statistical Analysis

Differences in group demographics were examined using independent sample T-tests. The effect of the intervention on walking (i.e., gait velocity and XcoM-BoS displacement) and standing-balance (i.e., ES, sway area, and total path) measures were analyzed using a three-way mixed-effects ANOVA with the Bonferroni correction, with the group (i.e., paper and smartphone) as the between-subjects factor. Testing condition (i.e., dual-task and single-task gait as well as eyes open or closed, visual surround, and force plate sway for standing tasks) and visit (i.e., pre-training, 4 weeks post-training, and 8-weeks post-training) were within-subject factors. A two-way mixed-effects ANOVA with the Bonferroni correction was utilized to investigate differences in the ABC, GDS, and TUG, with the group and visit as the between- and within-subject factors, respectively. Differences in the self-reported effectiveness, enjoyment, difficulty, and length of the exercise programs between groups were investigated using Independent-Samples Mann–Whitney U Tests. SPSS 28.0 (IBM Inc., Armonk, NY, USA) was used for all statistical analyses, with alpha levels set at 0.05.

## 4. Results

A total of 45 participants were screened for inclusion into the study. Following MOCA-blind testing, three participants were excluded due to scoring less than 18/22 on the assessment. An additional 11 individuals were excluded due to not meeting the study inclusion/exclusion criteria, or declining further participation in the study (Figure 3). A total of 31 older adults were invited to participate in the study and evaluated in the laboratory prior to being prescribed the 4-week balance-intervention program. Two adults dropped out of the study following the initial visit, both of whom were part of the smartphone intervention group. One individual did not begin the balance program and asked to be removed from the study following the first visit to the laboratory. The second individual began the intervention program, but during turning tasks performed during the second week of training, complained of dizziness and general feeling of uneasiness, and asked to dropout.

The remaining 29 participants reported completing all 12 sessions of intervention (Table 2). These older adults returned for their second and third visits to the laboratory on average 34.1 (3.0) days and 66.6 (4.8) days following their initial visit, respectively. Participants in the paper and smartphone groups spent approximately 45.0 (13.0) and 40.5 (10.1) min to complete each exercise session, respectively (*p* = 0.324). Due to technical issues, one participant’s phone did not record exercise timings.

### 4.1. Exercise Response

No group differences were found in the self-reported effectiveness (*p* = 0.621), enjoyment (*p* = 0.652), difficulty (*p* = 0.621), or length (*p* = 0.186) of the exercise program (Figure 4). Among adults randomized to the smartphone group, participants on average reported between slightly (score of 4 on range of 0–6) and quite true (score of 5 on a range of 0–6), in response to questions regarding whether the smartphone application allowed participants to accomplish tasks more quickly (4.4 ± 1.2), improved exercise performance (4.0 ± 1.1), made it easier to exercise (4.5 ± 0.9), and was useful in exercise (4.9 ± 1.0).

### 4.2. Clinical Measures

No group × visit interactions were found for the ABC (*p* = 0.199), GDS (*p* = 0.054), or TUG (*p* = 0.604; Table 3). Furthermore, no main effects of the group (*p* = 0.549, 0.991, 0.110) or visit (*p* = 0.417, 0.599, 0.640) were found for the measures of balance confidence, depression, and clinical balance, respectively.

### 4.3. Gait and Balance Measures

While a group×condition×visit interaction (*p* = 0.033) was found for step width, no pairwise differences were found between the smartphone and paper groups at any visit across any condition for step width. However, a 2.1 cm reduction in step width (*p* = 0.029) was demonstrated following the four-week intervention for the smartphone group during dual-task walking. No three-way (*p* = 0.314) or two-way interactions (*p* > 0.110) were found for gait velocity (Table 4). Furthermore, although no group main effect (*p* = 0.430) was found for gait velocity, condition (*p* < 0.001) and visit (*p* = 0.001) main effects were found. Participants demonstrated a 0.22 m/s slower walking speed during dual-task conditions, compared to single-task walking, and a 0.06m/s increase in gait velocity from pretraining to four-weeks post-training (*p* = 0.008).

Among the gait margin of stability measures, no group×condition×visit interaction was found for the XcoM-BoS displacement in the mediolateral (*p* = 0.810) and anteroposterior (*p* = 0.883) directions at toe-off. A visit×group interaction was found for the XcoM-BoS at toe-off in the mediolateral (*p* = 0.008), but not the anteroposterior (*p* = 0.613) direction. Pairwise comparisons revealed that the paper group increased the XcoM to BoS displacement at visit three in comparison to preintervention (*p* = 0.042) and 4-weeks postintervention (0.037). Although no other two-way interactions were demonstrated, a main effect of condition (*p* < 0.001) was demonstrated for the XcoM-BoS at toe-off in both the mediolateral and anteroposterior directions, with participants maintaining the XcoM closer to the BoS during the more demanding dual-task conditions.

For standing outcomes, a group×condition×visit interaction was found for the total excursion (*p* = 0.003; Table 5), with an approximately 0.89cm decrease in the excursion demonstrated from preintervention to 4 weeks postintervention in the smartphone group when performing the most demanding standing task of both a ground and visual sway reference (*p* = 0.027). No group×condition×visit interaction was demonstrated for the ES (*p* = 0.515) and sway area (*p* = 0.530). There was, however, a significant main effect of condition for the ES (*p* < 0.001) and sway area (*p* < 0.001), with increased anteroposterior movement and a greater sway area demonstrated with increased difficulty of the postural task.

## 5. Discussion

Results of this study demonstrate the feasibility and effectiveness of delivering a home-based balance intervention to older adults via either smartphone or paper. In support of our first hypothesis, smartphone-delivered interventions were successfully performed by older adults based on compliance, fidelity, and the time required to complete. This study had a high retention rate, with 29/31 (93.5%) participants completing all three visits to the laboratory. Of the two participants who dropped out, smartphone usage was not reported as the deciding factor, but rather a lack of motivation to begin the exercise program and the onset of dizziness. Adherence results were extremely high, with all 29 nondropout participants reporting 100% completion of the 12 exercise sessions, which is greater than the 70% cut-off point often used to signify sufficient participation [25,26]. It is possible that participant retention and adherence were at a high level in the current study due to: (1) the participants receiving a weekly phone call to ensure safety and compliance from the trainer; (2) the increased interest among our cohort of older adults; (3) the short duration of the balance intervention prescribed. Older adults recruited were also highly motivated, active, and educated, with participants reporting enjoyment of the program and, on average, at least 4 years of higher education. Such levels of adherence and compliance are likely not sustainable among the general population. While exercise adherence is often a significant hindrance for older adults, multiple factors can affect the rate, such as socioeconomic status, education level, living arrangements, health status, physical fitness, and depression [27].

Participants in this study completed all 12 sessions of exercise within approximately 5 weeks of receiving training. Furthermore, no differences were reported in the amount of time spent in each session or in the enjoyment and difficulty of the exercise programs between groups, with participants in the smartphone group indicating positive usage of technology to perform exercise. While technology is often indicated as a barrier, adoption rates have been steadily increasing, with 77% of older adults in the United States indicating adoption of smartphones by 2016 [28,29]. In the current study, prior smartphone usage was high, with participants indicating greater than 8.9 years of ownership at enrollment. However, while some participants indicated requiring “more time to get oriented to the smartphone application”, many provided feedback regarding greater motivation and encouragement to perform the exercises when prompted by the phone. Familiarity of phone usage and technological literacy has previously been shown to facilitate technology adoption in older adults [30], however greater education and training might be required if such programs were to be delivered to adults with reduced exposure to smartphones, which might affect both enjoyment and compliance.

In partial support of our second hypothesis, participants demonstrated increased gait velocity, reduced step width during the dual-task condition among the smartphone participants, and increased XcoM-BoS displacement within the paper group. Furthermore, the smartphone group demonstrated reduced total path excursion during the most difficult standing condition following the four-week intervention. Recent reviews suggest that balance training programs yield optimal improvements in static and dynamic balance if they are delivered in a supervised setting [31] and over the course of 11–12 weeks, three times per week, for 31–45 min per session [32]. Results from the present investigation however, suggest that slight improvement in gait performance can be seen with an unsupervised four-week balance intervention over twelve sessions. While Wongcharoen and colleagues demonstrated reduced gait speed under narrow walking conditions among participants who received a similar 4-week balance training program [11], our study indicates that balance training might slightly increase speed during single- and dual-task level walking. A gait velocity increase of 0.06 m/s, as found in our older adult participants, has previously been shown to be of small meaningful change [33]. Furthermore, the reduced step width in the smartphone group and increased margin of stability in the mediolateral direction among the paper group were indicative of altered strategies to maintain balance during the more demanding dual-task walking conditions. While some training tasks provided include motor–motor exercises, only motor–cognitive dual-task conditions were evaluated during each participant’s visit to the laboratory. Future evaluations should include: (1) motor–motor dual-tasks to evaluate the efficacy of task-specific training; (2) motor–cognitive exercise tasks, as an impaired ability to simultaneously perform cognitive tasks while maintaining balance is associated with increased risk of recurrent falls [34]. Findings from this study, however, demonstrate the efficacy and feasibility of the current four-week program.

In partial support of our third hypothesis, individuals demonstrated no difference in gait and standing-balance performance during the second and third testing sessions. A greater number of participants, a longer retention period, and analysis of both adults with and without a history of falls should be investigated further in order to better understand retention effects. Prior retention of balance following a five-week group-based balance-training program has demonstrated mixed results among older adults up to five weeks postintervention [35]. Seidler and colleagues indicated that those adults who reported a history of falls had an impaired acquisition of long-lived improvement in balance performance [35]. A greater number of assessments throughout the intervention and postintervention period might also be beneficial, as intrasubject variability and single-timepoint measurements might not be robust to changes in performance. Some participants in the current student complained of fatigue or other daily life complications during some laboratory testing sessions, which might further affect single-timepoint assessments. Prior results have demonstrated that an older adult’s typical gait in the home environment cannot be reliably measured from a single evaluation in the laboratory [36]; therefore, methods for easily assessing adults in the home environment, along with delivering home-based interventions, might be beneficial and should be pursued in subsequent studies.

Although this study demonstrates the feasibility and efficacy of a home-based balance program delivered by smartphone or paper, there are a few limitations. First, some participants (three pairs) who were recruited joined as a husband/wife pair. Randomization was set such that both spouses were assigned to the same group in order to ensure blinding. Although testing and training were performed on an individual basis, it is possible that compliance and adherence increased due to peer motivation among these participants. Second, different methods for measuring the time for exercise completion may have introduced errors in participant reporting among the paper group. Given that these participants self-reported their time to complete each exercise session, there could have been an over- or underestimation of this measure. However, there is reason to believe the veracity of the data, as weekly check-in calls consistently provided reports of sessions taking more than 30 min to complete. Third, although we had sufficient power to detect changes in the primary outcome measures, additional participants may elucidate the training effect on other measures. Furthermore, all participants recruited were white, highly educated, and active members in the community. While it is unclear how generalizable the findings will be to a more diverse group of older adults, the benefits of home-based delivery of intervention should continue to be investigated. Future studies will examine the efficacy of single- and dual-task home-based training programs in more underserved and underrepresented communities.

## Figures and Tables

**Figure 1 sensors-23-07451-f001:**
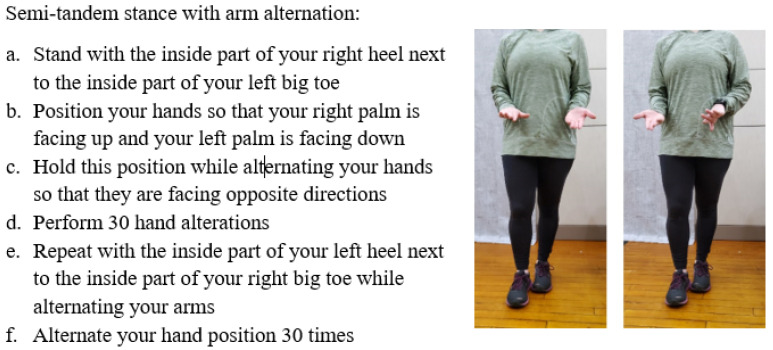
Example exercise instruction set with images provided to participants in the paper-based intervention group.

**Figure 2 sensors-23-07451-f002:**
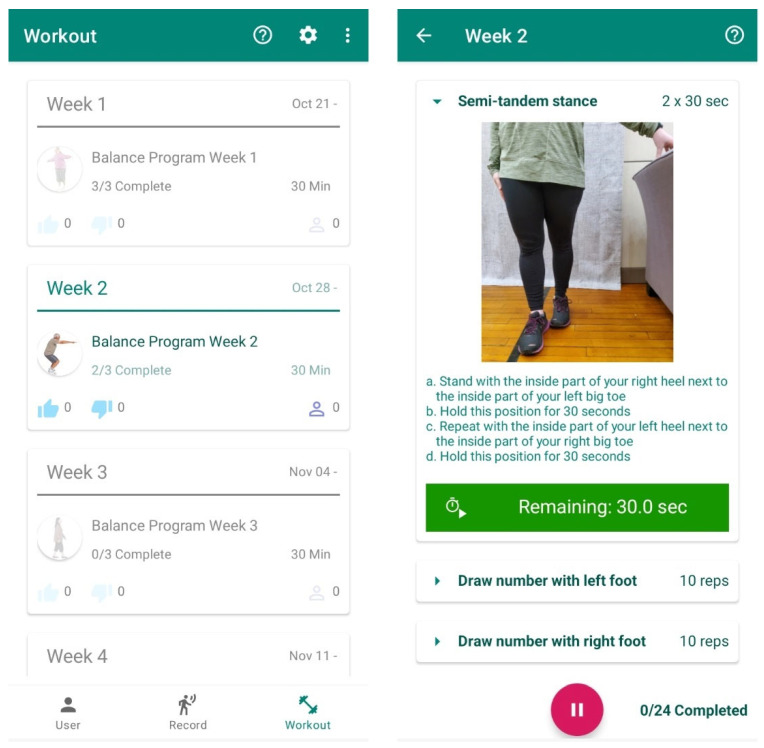
Weekly workout programs and example exercise instruction set with images as displayed to participants in the smartphone-based intervention group.

**Figure 3 sensors-23-07451-f003:**
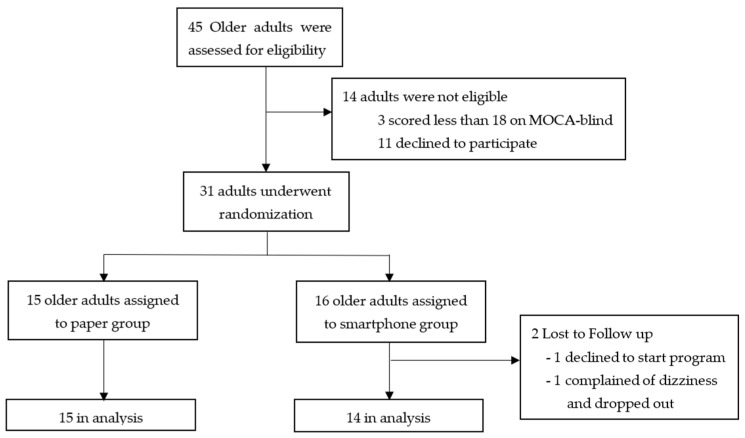
Randomized controlled trial study design.

**Figure 4 sensors-23-07451-f004:**
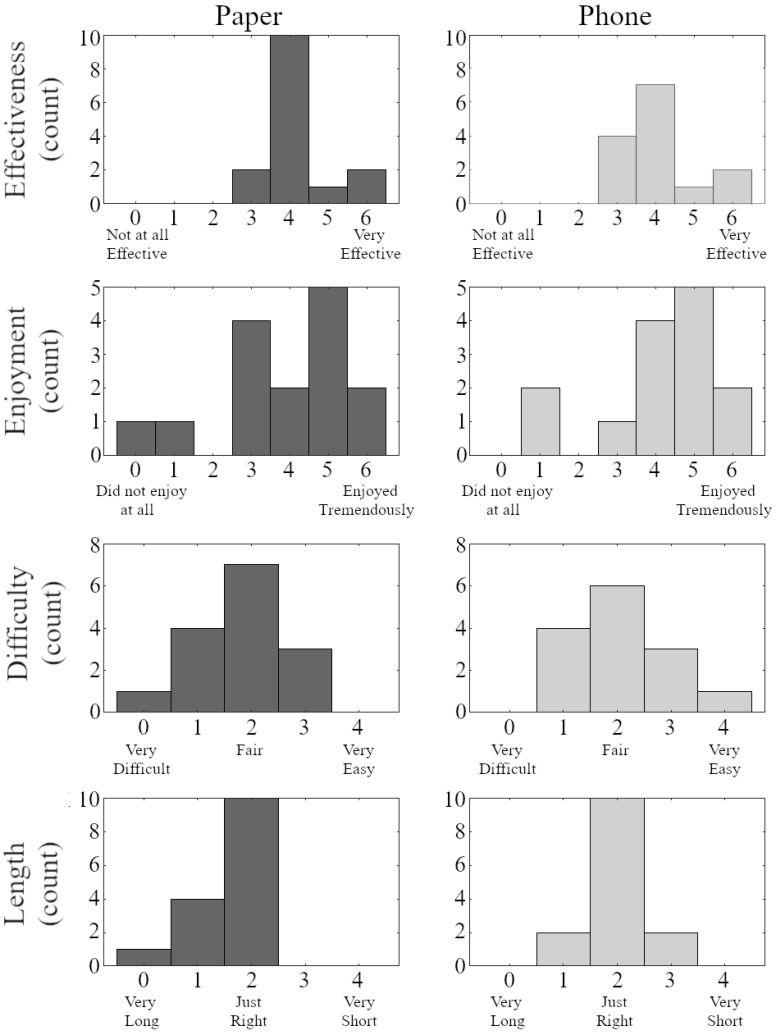
Self-reported effectiveness and enjoyment of the exercise program on a scale of 0–6, as well as exercise difficulty and length on a scale of 0–4, for all participants in the study following the four-week intervention program.

**Table 1 sensors-23-07451-t001:** Exercise schedule provided to older adults in both intervention groups. Of the 12 exercises prescribed for each week, older adults were presented with images, detailed instruction, as well as the duration and number of repetitions for each activity through the smartphone application or paper instruction set. Participants were further asked to complete all 12 exercises twice during each session, for a total of 24 exercises completed.

Tasks	Week 1	Week 2	Week 3	Week 4
Stance Activities No manipulation	1. Feet together stance 2. Knee marching 3. Stepping in different directions 4. Standing with forwards and backwards straight leg swings 5. Standing with sideways straight leg swings	1. Semi-tandem stance 2. Draw number with left foot 3. Draw number with right foot	1. Tightrope standing	
Hand manipulation	6. Standing with feet together and reach as far and as safely as you can in different directions	4. Semi-tandem stance with arm alternation 5. Knee marching with arm alternation 6. Standing knee lifts to each hand	2. Tightrope standing with reaching as far and as safely as you can in different directions 3. Knee marching with clap hand under knee	1. Tightrope standing with reaching as far and as safely as you can in different direction 2. Standing on toes with arms lift overhead 3. Knee lifts while standing with raise the opposite arm overhead
Transitional Activities	7. Sit to stand and walk in a circle 8. Sit to stand and walk in a figure eight	7. Sit to stand and cross-legged walk 8. Sit to stand and walk with high step	4. Sit to stand and walking while knee marching in a figure eight	4. Stand up and cross leg walk in a figure of eight
Gait Activities No manipulation	9. Narrow walking 10. Walking with knee marching in a figure eight 11. Toe walking	9. Narrow walking 10. Cross leg walking in a figure eight 11. Walk around obstacles–place two items on floor	5 Narrow walking 6.Tightrope walking 7. Backward long-step walking 8. Backward tightrope walking 9. Sideways walking with step across in front of other leg 10. Sideways walking with step across and behind other leg	5. Narrow walking 6. Sideways walking with step across in front of/behind other leg 7. Walk up and down stairs
Hand manipulation	12. Toe walking with arm alternation	12. Walk around obstacles with arm alternation	11. Tightrope walking while carrying bag 12. High step walking with clap hand under knee	8. Tightrope walking with arm alternation 9. Backward tightrope walking with arm alternation 10. Backward long-step walking with arm alternation 11. Backward toe walking with arm alternation 12. Backward high step walking with hand clap under knee

**Table 2 sensors-23-07451-t002:** Participant demographics (mean (SD)).

	Smartphone Group (*n* = 14)	Paper Group (*n* = 15)	*p*-Value
Female	9	10	
Reported a fall in past year	6	9	
Age (years)	75.6 (8.9)	78.2 (8.4)	0.420
Height (m)	1.64 (0.06)	1.61 (0.11)	0.175
Weight (kg)	69.4 (14.9)	68.5 (16.6)	0.443
Number of medications	6.0 (3.4)	5.1 (3.4)	0.238
Years of smartphone use	7.1 (5.1)	9.7 (5.6)	0.106
Education level (years)	16.4 (2.3)	16.5 (2.3)	0.898

**Table 3 sensors-23-07451-t003:** Clinical performance across time among older adults in smartphone and paper intervention groups.

	Smartphone (*n* = 14)			Paper (*n* = 15)		
	Visit 1	Visit 2	Visit 3	Visit 1	Visit 2	Visit 3
ABC (%)	88.3 (7.8)	89.5 (7.4)	89.0 (9.2)	87.9 (10.8)	84.0 (15.8)	88.4 (10.7)
GDS (n/14)	1.07 (1.38)	1.00 (1.57)	1.21 (1.72)	1.20 (1.57)	1.27 (1.91)	0.80 (1.42)
TUG (sec)	10.7 (2.0)	10.6 (1.7)	10.8 (2.0)	12.7 (3.5)	12.3 (3.8)	12.2 (3.5)

**Table 4 sensors-23-07451-t004:** Gait spatiotemporal and balance performance under single- and dual-task conditions.

	Smartphone (*n* = 14)			Paper (*n* = 15)		
	Visit 1	Visit 2	Visit 3	Visit 1	Visit 2	Visit 3
Gait Velocity (m/s) ^†‡^						
Single-Task	1.02 (0.19)	1.06 (0.20)	1.05 (0.18)	0.97 (0.22)	0.95 (0.24)	0.98 (0.23)
Dual-Task	0.77 (0.18)	0.77 (0.27)	0.81 (0.25)	0.72 (0.18)	0.75 (0.20)	0.81 (0.21)
Step Width (cm) *						
Single-Task	9.2 (4.6) ^a^	9.9 (3.8)	9.4 (4.0)	10.7 (2.0)	10.1 (3.3)	10.5 (3.6)
Dual-Task	11.9 (3.5) ^a,b^	9.8 (3.9) ^b^	10.4 (3.5)	11.1 (3.7)	10.9 (4.1)	12.3 (3.9)
XcoM-BOS at TO—Anteroposterior (cm) ‡						
Single-Task	19.1 (5.9)	19.7 (5.4)	19.4 (5.1)	16.4 (7.9)	16.8 (8.6)	17.8 (8.7)
Dual-Task	10.2 (5.7)	10.4 (7.6)	11.6 (7.3)	9.6 (6.9)	10.1 (7.6)	11.8 (7.3)
XcoM-BOS at TO—Mediolateral (cm) ^§ ‡^						
Single-Task	19.1 (3.2)	18.4 (3.0)	18.3 (2.6)	16.9 (5.0)	17.2 (5.3)	17.7 (5.4)
Dual-Task	17.9 (2.7)	17.3 (2.9)	17.5 (2.7)	16.4 (4.4)	16.4 (4.9)	17.3 (5.0)

* Group×condition×visit interaction; The letters ^a,b^ indicate pairwise differences (*p* < 0.05); ^§^ Visit × group interaction; ^†^ Visit main effect (*p* < 0.05); ^‡^ Condition main effect (*p* < 0.05).

**Table 5 sensors-23-07451-t005:** Performance during the sensory organization test for participants in the smartphone and paper intervention groups.

	Smartphone (*n* = 14)			Paper (*n* = 15)		
	Visit 1	Visit 2	Visit 3	Visit 1	Visit 2	Visit 3
Equilibrium Score (%) ^‡^						
EO	92.7 (2.3)	93.1 (2.3)	92.0 (2.8)	92.2 (2.3)	91.4 (2.6)	91.9 (2.8)
EC	91.9 (3.1)	91.2 (3.4)	91.0 (3.8)	90.9 (3.2)	89.9 (4.1)	88.7 (5.7)
VS	89.6 (6.3)	90.0 (4.4)	91.5 (3.7)	89.2 (6.8)	88.6 (5.2)	89.0 (5.0)
EO Sway	71.5 (13.3)	72.5 (11.8)	75.0 (9.5)	70.9 (15.4)	73.1 (11.5)	74.3 (10.1)
EC Sway	60.5 (24.6)	54.5 (22.2)	68.3 (17.7)	56.3 (20.6)	58.6 (22.6)	60.1 (21.8)
VS Sway	53.1 (26.9)	54.0 (18.0)	62.4 (11.4)	48.7 (19.8)	54.9 (20.6)	55.3 (21.7)
Total Excursion *						
EO	2.29 (0.52)	2.29 (0.50)	2.32 (0.48)	2.30 (0.48)	2.33 (0.50)	2.38 (0.49)
EC	2.32 (0.48)	2.33 (0.47)	2.34 (0.47)	2.36 (0.47)	2.35 (0.51)	2.54 (0.72)
VS	2.43 (0.44)	2.37 (0.57)	2.39 (0.45)	2.48 (0.52)	2.53 (0.56)	2.59 (0.63)
EO Sway	2.83 (0.52)	2.92 (0.98)	2.77 (0.64)	2.83 (0.53)	2.84 (0.52)	2.98 (0.59)
EC Sway	2.89 (0.47)	2.96 (0.91)	2.64 (0.50)	3.12 (0.69)	3.14 (0.96)	3.24 (1.35)
VS Sway	3.28 (0.90) ^b^	2.39 (0.71) ^a,b^	2.63 (1.17)	3.10 (1.34)	3.42 (1.38) ^a^	3.34 (1.48)
Sway Area (cm^2^) ^‡^						
EO	6.77 (3.73)	7.10 (2.39)	7.04 (3.51)	8.19 (5.25)	7.41 (4.21)	7.18 (5.82)
EC	6.96 (2.57)	7.99 (3.50)	7.90 (3.42)	8.32 (5.14)	7.06 (3.72)	7.42 (5.30)
VS	7.08 (2.26)	7.80 (3.38)	7.79 (3.00)	9.87 (7.54)	8.22 (4.84)	7.69 (5.14)
EO Sway	9.56 (3.08)	11.51 (7.40)	9.64 (4.28)	10.08 (5.24)	10.65 (5.55)	9.53 (4.28)
EC Sway	12.20 (7.05)	16.17 (13.73)	12.02 (8.59)	15.51 (11.17)	14.52 (6.53)	14.07 (6.94)
VS Sway	16.11 (9.40)	22.74 (29.04)	16.72 (11.16)	23.75 (13.23)	19.34 (10.11)	16.44 (9.81)

* Group×condition×visit interaction; The letters ^a,b^ indicate pairwise differences (*p* < 0.05); ^‡^ Condition main effect (*p* < 0.05); EO: eyes open; EC: eyes closed; VS: visual sway referenced; Sway: ground sway referenced.

## Data Availability

The data presented in this study are available on request from the corresponding author. The data are not publicly available due to potentially identifying information within the datasets.

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
