# Peer review of "Smartphone- and Paper-Based Delivery of Balance Intervention for Older Adults Are Equally Effective, Enjoyable, and of High Fidelity: A Randomized Controlled Trial"

_sensors, 2023, doi:10.3390/s23177451_

Round 1

Reviewer 1 Report

The manuscript is well-written and organized. The study presents an high scientific soundness in the field of rehabilitation.

Here are few suggestions for the Authors:

- line 222-225: the Authors, regarding the table 2, have reported that "On average, participants in the paper and smartphone groups spent approximately 45.0 (13.0) and 40.5 (10.1) minutes to complete each exercise session, respectively (P = 0.324)", but I can't see these informations in the Table 2.

Moreover, if the Authors took in consideration the minutes that the subjects spent to complete the training session, it could be preferible that in the "intervention" section is explained if all the training sessions are carried out in the same time of the day. This is as the muscle fatigue could variate during the day, especially in older adults.

- Table 1 and Figure 3: It is preferible to modify the layout as part of the test appears missing

- Table 2: As the intact cognitive function (based on scoring 18/22 or greater on the Montreal Cognitive Assessment – Blind) is part of the inclusion criteria, the Authors could include this information about the demographics. 

Moreover, in the discussion the Authors have highlighted that the adherenceto the study was at high level due also to the education of the participants. The Authors could also include the education (years) as another demographic data to support this part of the discussion.

Reviewer 2 Report

The manuscript presents a small interesting study on the analysis of a home balance intervention. The topic is well introduced, presented, and discussed.

Only a few hints and comments are open to improve the manuscript. The manuscript is acceptable.

Format:

·         Using keywords different from the terms in the title would increase accessibility.

·         Tables 1, 3, 4, and 5 are not presented in total.

Discussion:

·         The limitation of different methods to measure the time for exercise performance was not mentioned.

Round 2

Reviewer 1 Report

The manuscript was modified according to the suggestions. Thanks to the Authors 

Author Response

The authors would like to thank the reviewer for their time and effort in reviewing the manuscript.